# Proteins associated with neutrophil degranulation are upregulated in nasopharyngeal swabs from SARS-CoV-2 patients

Emel Akgun[1,2], Mete Bora Tuzuner[2], Betul Sahin[2], Meltem Kilercik[1,2], Canan Kulah[2], Hacer Nur Cakiroglu[2], Mustafa Serteser [1,2], Ibrahim Unsal[2], Ahmet Tarik Baykal [1,2] *

1 Department of Medical Biochemistry, Faculty of Medicine, Acibadem University, Istanbul, Turkey,
2 Acibadem Labmed Clinical Laboratories, Istanbul, Turkey

* ahmet.baykal@acibadem.edu.tr

## Abstract

COVID-19 or severe acute respiratory syndrome coronavirus 2 (SARS-CoV-2) appeared throughout the World and currently affected more than 9 million people and caused the death of around 470,000 patients. The novel strain of the coronavirus disease is transmittable at a devastating rate with a high rate of severe hospitalization even more so for the elderly population. Naso-oro-pharyngeal swab samples as the first step towards detecting suspected infection of SARS-CoV-2 provides a non-invasive method for PCR testing at a high confidence rate. Furthermore, proteomics analysis of PCR positive and negative naso-oropharyngeal samples provides information on the molecular level which highlights disease pathology. Samples from 15 PCR positive cases and 15 PCR negative cases were analyzed with nanoLC-MS/MS to identify the differentially expressed proteins. Proteomic analyses identified 207 proteins across the sample set and 17 of them were statistically significant. Protein-protein interaction analyses emphasized pathways like Neutrophil degranulation, Innate Immune System, Antimicrobial Peptides. Neutrophil Elastase (ELANE), Azurocidin (AZU1), Myeloperoxidase (MPO), Myeloblastin (PRTN3), Cathepsin G (CTSG) and Transcobalamine-1 (TCN1) were found to be significantly altered in naso-oropharyngeal samples of SARS-CoV-2 patients. The identified proteins are linked to alteration in the innate immune system specifically via neutrophil degranulation and NETosis.

## 1. Introduction

Severe acute respiratory distress syndrome-associated coronavirus-2 (SARS-CoV-2) also known as COVID-19 first appeared in Wuhan, Hubei, China in December 2019, and now it is widespread around the World. The replication of the virus is through the human airway epithelial cells where it targets the receptors of human Angiotensin-Converting Enzyme 2 (ACE-2). The high mortality observed in COVID-19 is associated with severe acute respiratory distress and systemic coagulopathy. Many COVID-19 patients show a postponed onset of respiratory issues but then develop into more severe situations. We wanted to investigate the

**Funding:** Acibadem Labmed Clinical Laboratory provided support for this study in the form of salaries for MBT, BS, CK, HNC, and IU. The specific roles of these authors are articulated in the 'author contributions' section. Data acquisition was also conducted in Acibadem Labmed Clinical Laboratory's proteomics facility. The funders had no further role in study design, data collection and analysis, decision to publish, or preparation of the manuscript. The authors received no other specific funding for this work.

**Competing interests:** The authors have read the journal's policy and have the following competing interests: MBT, BS, CK, HNC, and IU are employees of Acibadem Labmed Clinical Laboratory. This does not alter our adherence to PLOS ONE policies on sharing data and materials. There are no patents, products in development or marketed products associated with this research to declare.

**Abbreviations:** CE, collision energy; DIA, data-independent acquisition; FA, formic acid; UPLC, Ultra performance liquid chromatography; UPX, Universal protein extraction kit.

molecular changes in the COVID-19 patients' naso-oropharyngeal swab samples via comparison to the proteome of PCR negative cases. Our goal is to find pathways associated with the site of infection through proteomics analysis. 17 statistically significant protein alterations lead us specifically to neutrophil degranulation pathways.

During airway infections, Neutrophils are the first wave of defense that also defines the disease outcome. Neutrophils have multiple functions in viral infections such as inactivation of the virus, achieved by phagocytosis, ROS production, proteolytic enzyme release, NET activation (NETosis, Neutrophil Extracellular Traps). Neutrophils also interact with immune cells and secreting cytokines participate in eliciting an antiviral response.

Neutrophil granulocytes express several enzymes linked to controlling host infection. As the cytotoxic molecules are released from the granules they have an impact on the inflammatory response. Such adverse molecules cause severe damage to the host where it exhibits itself as perivascular infiltrates around the capillaries in the lungs as observed in SARS-CoV-2 patients. The identified up-regulated proteins Myeloperoxidase, Myeloblastin, Neutrophil Elastase, Cathepsin G, and Azurocidin (MPO, PRTN3, ELANE, CTSG, and AZU1) in naso-oropharyngeal swab samples are discussed to highlight the molecular mechanism changes in the site of infection.

## 2. Materials and methods

### 2.1. Study population and sample collection

Infected and non-infected study groups were formed from the patients who applied to Acıbadem Health Group hospitals with suspected SARS-CoV-2 infection. Cases were diagnosed on the basis of the interim guidance of the World Health Organization (WHO) (World Health Organization; Geneva: 2019. Clinical Management of Severe Acute Respiratory Infection When Middle East Respiratory Syndrome Coronavirus (MERS-CoV) Infection is Suspected: Interim Guidance.) and diagnosis and treatment guidelines of COVID-19 in Turkey (COVID-19 (SARS-CoV-2 INFECTION) GUIDE Republic of Turkey, Ministry of Health, April 14th 2020, Ankara). Selected patients did not have any other known infections at the time of diagnosis. A total of 30 patients were enrolled in the study: 15 patients PCR positive for SARS-CoV-2 (mean age: 38.9±13.8) and 15 patients PCR negative for SARS-CoV-2 (mean age: 36.6 ±15.9). Infected group was equally distributed regarding gender. Most of the non-infected patients were men (60%). As a routine application of our clinical laboratory, WBC count of the patients were also performed (Neutrophil count x$10^9$/L mean: 0.6±0.5 for non-infected and 5.6±10.2 for infected).

Samples were collected using a nasal and oropharyngeal (NUCLISWAB) swab (Salubris, Turkey) and were placed in a tube with Universal Transport Media. After collection the samples were stored at -80˚C before proteomics analysis. No minors were included in the study and a written informed consent was obtained from each patient that was enrolled in the study. Ethical approval for the conduct of the study was given by Acibadem Mehmet Ali Aydinlar University Human Scientific and Ethical Review Committee (Approval ID: 2020-07/9).

### 2.2. COVID-19 RT-PCR test

For molecular testing of SARS-CoV-2; the extraction of nucleic acids from the samples was performed by a manual liquid phase method using Bio-Speedy Nucleic Acid Isolation Kit (Bioeksen, Turkey). Nucleic acid amplification test (NAT) was carried out by Bio-Speedy COVID-19 RT-qPCR Detection Kit (Bioeksen, Turkey) according to the manufacturer's instructions on RotorGene (Qiagen, Germany) Real-Time PCR instrument. The test briefly

achieves a one-step reverse transcription (RT) and real-time PCR (qPCR) targeting SARS--CoV-2 specific RdRp (RNA-dependent RNA polymerase) gene fragment.

### 2.3. LC-MSMS analysis

A shorter sample preparation approach was applied to obtain the tryptic peptides for analyses. 50 ul of the naso-oropharyngeal transport solution was taken and lyophilized. The powder was reconstituted in 20ul of 50 mM Ammonium bicarbonate solution with 1 ul of protease inhibitor mixture (Thermo Scientific). The mixture was centrifuged at 14000 xg for 10 min and the supernatant was transferred to a clean Eppendorf tube. DTT was added to 10 mM final concentration and incubated at 60˚C for 10 min. The mixture was alkylated in dark for 30 min with 20mM IAA. To the resulting mixture 1 ug of sequencing grade trypsin (Promega Gold) was added and incubated at 55˚C for 1.5 hrs. The digest was acidified with 1 ul formic acid and transferred to an LC vial for injection. The samples were analyzed by the protocols in our previous studies [1]. Briefly, tryptic peptides were trapped on a Symmetry C18 (5μm,180μm i.d. × 20 mm) column and eluted with ACN gradient (4% to 40% ACN, 0.3 ul/min flow rate) with a total run time of 60 min on a CSH C18 (1.7 μm, 75 μm i.d. × 250 mm) analytical nano column. Data were collected in positive ion sensitivity mode using a novel data-independent acquisition mode (DIA) coined as SONAR [2] with a quadrupole transmission width of 24 Da. Progenesis-QIP (V.2.4 Waters) was used for data processing.

### 2.4. PPI network and pathway analysis

Protein-protein interaction networks functional enrichment analysis was carried out using the STRING (http://string-db.org/, v11.0) database with the highest confidence interaction score level to identify possible pathways related to the identified proteins. Textmining, experiments, databases co-expression, neighborhood, gene fusion, and co-occurrence selected as active interaction sources. The minimum interaction score was set to high (high confidence = 0.700). REACTOME (http://www.reactome.org) pathways analysis tool was also used for processing.

## 3. Results

### 3.1. Label-free proteomics

Proteins were extracted from a small amount of naso-oropharyngeal samples, fast tryptic digestion was applied and followed with a 60 min reverse-phase separation. NanoLC-MSMS analysis provided the identification of 207 protein groups with high confidence (<1% FDR) (S1 and S2 Tables). Statistical analysis done in Progenesis QIP software identified 17 proteins to be statistically significantly expressed in patients' naso-oropharyngeal samples (Table 1).

### 3.2. Bioinformatic analysis

Pathway analysis of the significantly altered protein levels between COVID-19 positive and negative patients' naso-oropharyngeal swab samples were analyzed using the STRING online database. The PPI network obtained, contained 17 differentially expressed proteins with 15 nodes (disconnected nodes were not shown) and 14 edges as shown in Fig 1A. The main cluster includes Neutrophil Elastase (ELANE), Azurocidin (AZU1), Myeloperoxidase (MPO), Myeloblastin (PRTN3), Cathepsin G (CTSG) and Transcobalamine-1 (TCN1). The abundance of these proteins were found to be increased in COVID-19 positive patient samples compared to negative ones. REACTOME pathway enrichment analyses of the differentially expressed proteins were performed. Nine of the proteins were primarily associated with the immune system pathway. Proteins clustered in the PPI network were mainly enriched in the neutrophil

**Table 1. Significantly altered protein identification list.**

| Accession | Unique peptides | *P value* | Fold change (Pos/Neg) | Description |
|---|---|---|---|---|
| Q9BQE3 | 2 | 0.0234 | 0.41 | Tubulin alpha-1C chain |
| P20160 | 4 | 0.0413 | 2.03 | Azurocidin |
| P01876 | 12 | 0.0073 | 2.04 | Immunoglobulin heavy constant alpha 1 |
| P29401 | 2 | 0.0337 | 2.10 | Transketolase |
| P09104 | 2 | 0.0075 | 2.11 | Gamma-enolase |
| P20061 | 3 | 0.0294 | 2.48 | Transcobalamin-1 |
| P04004 | 2 | 0.0001 | 2.65 | Vitronectin |
| P02790 | 7 | 0.0000 | 2.71 | Hemopexin |
| P0DOX7 | 3 | 0.0127 | 2.83 | Immunoglobulin kappa light chain |
| P08246 | 2 | 0.0035 | 2.91 | Neutrophil elastase |
| Q9UKL4 | 2 | 0.0011 | 3.18 | Gap junction delta-2 protein |
| Q16695 | 2 | 0.0003 | 3.20 | Histone H3.1t |
| P01871 | 2 | 0.0003 | 3.44 | Immunoglobulin heavy constant mu |
| P08311 | 3 | 0.0097 | 3.67 | Cathepsin G |
| P05164 | 9 | 0.0050 | 3.72 | Myeloperoxidase |
| P00450 | 4 | 0.0020 | 5.06 | Ceruloplasmin |
| P24158 | 2 | 0.0023 | 29.42 | Myeloblastin |

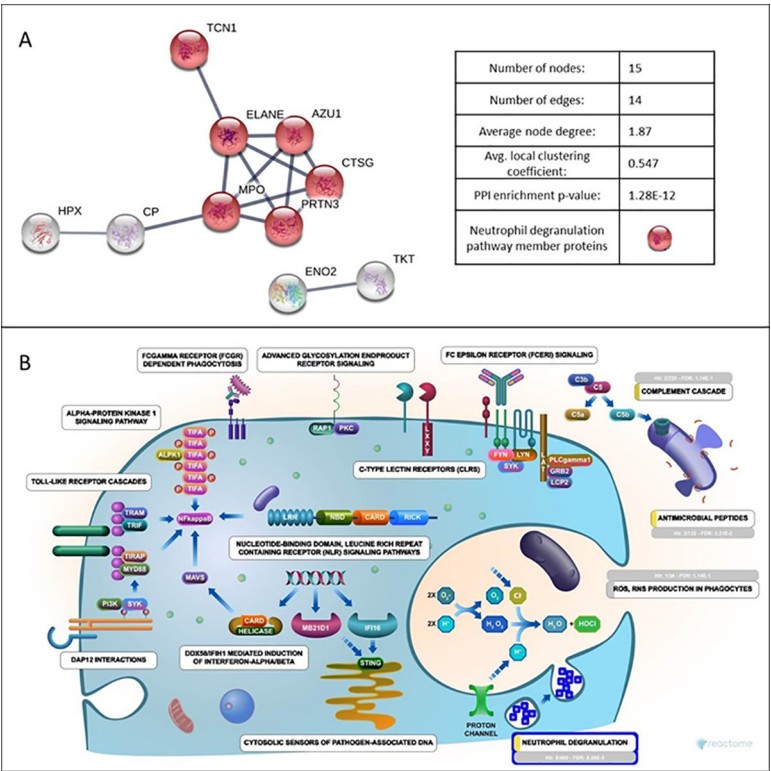

**Fig 1. STRING and Reactome analysis of identified proteins.** A-Protein-protein interaction network. B-Diagram of Initiate Immune System pathway (HSA-168249, FDR = 2.3E-5) from Reactome, showing a significant enrichment for neutrophil degranulation pathway (HSA-6798695, FDR = 2.01E-7) having the most hits.

degranulation pathway (HSA-6798695, FDR = 2.01E-7) which is a subpathway of the innate immune system (Fig 1B).

## 4. Discussion

Neutrophils play a vital role in airway infections and may also define the disease outcome and carry out various functions in viral infections ranging from phagocytosis, degranulation to the generation of neutrophil extracellular traps (NETs) [3]. Neutrophils also interact with the immune cells and secreting cytokines participate in eliciting the antiviral response. Degranulation mechanism is part of the innate immune system which is necessary for the fight against infection and provides the neutrophils with the necessary tools. A recent study reported the accumulation of neutrophils in severe COVID-19 patients compared to non-severe patients via analyzing 6 different studies [4]. We also observed the same profile among our SARS-CoV-2 patients where the neutrophils were exceedingly high compared to the healthy group (Fig 2). In addition, our data led us to significant indications that the dysregulation of degranulation and NETosis mechanisms may also be involved in SARS-CoV-2 infection. In SARS-CoV-2 patients' naso-oropharyngeal samples, we have identified azurophilic granule (AG) proteins like Myeloperoxidase (MPO), elastase (ELANE), cathepsin G (CTSG), azurocidin 1 (AZU1) and proteinase 3 (PRTN3) to be highly overexpressed.

### 4.1. SARS-CoV-2 associated dysregulation in neutrophil degranulation

It is known that neutrophils are the first line of defense against the onset of a viral infection and they begin the process of defending against microorganisms by releasing antiviral enzymes and toxins stored in their granule. These azurophilic neutrophils undergo limited exocytosis when activated [5] and their primary role is believed to be killing and degradation of engulfed microbes in the phagolysosome [6]. Therefore our observation of upregulated proteins related

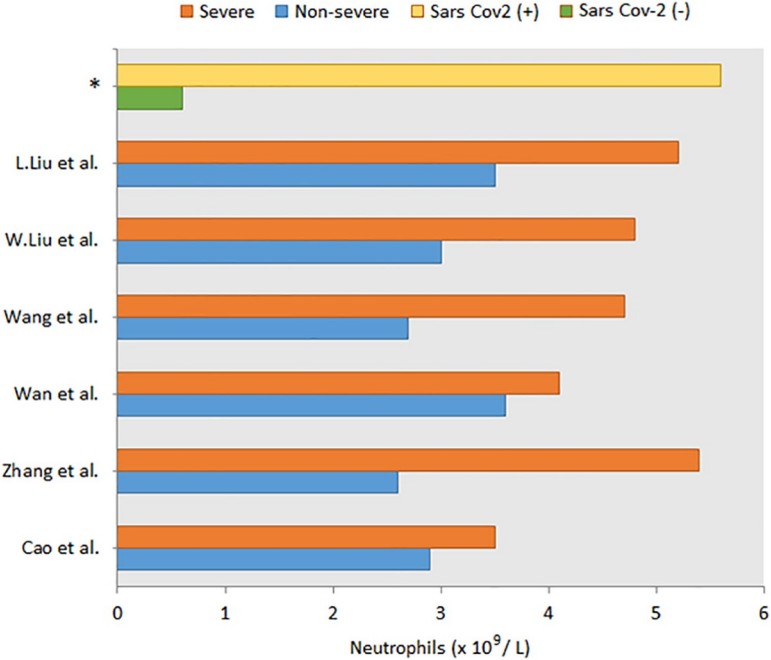

**Fig 2. Neutrophil counts between severe and non-severe SARS-CoV-2 (+) groups.** (*) indicates the findings of the current study. (ref eklenecek).

to neutrophil degranulation in SARS-CoV-2 patients' naso-oropharyngeal samples is not suprising. Although neutrophil degranulation having a positive impact on clearance of SARS-CoV-2 from the nasopharynx, there is still a debate in the inflammatory microenvironment the process is dysregulated and may lead to tissue damage by the secreted molecules from the granules [7].

Myeloperoxidase (MPO), is the most abundant protein in neutrophils and represents 5% of their total protein content [8]. Lau et al. reported the association of MPO and neutrophil activation via CD11b/CD18 integrins which is an indicator of MPO's possible contribution to neutrophil recruitment to the site of inflammation [9]. Our data supports this hypothesis for SARS-CoV-2 patients which we detected significantly high expressions of MPO ($\approx$4 fold, $p < 0.05$) at the nasopharynx region. Furthermore it is tempting to speculate this neutrophil burst and even more overexpressed MPO may cause production of excess hypochlorous acid (HOCl) and other reactive oxidants that also damages the nasopharynx tissue. So MPO may be an important factor where the protective inflammation can become pathological in SARS-CoV-2 cases.

The resolution phase of inflammation is essential to curtail inflammation and restore tissue homeostasis [10]. Neutrophil apoptosis and the cytokines released from macrophages during phagocytosis of the neutrophils are necessary in the resolution of inflammation [11]. The resolution of inflammation and tissue repair processes are aided by the cytokines released from macrophages during phagocytosis of the neutrophils. The dysregulation in the macrophage orchestrated phagocytosis and neutrophil apoptosis mechanisms leads to inflammation [12]. PRTN3, another azurophilic granule (AG) protein that we identified, is a serine protease enzyme that is involved in granulocyte differentiation and expressed in the neutrophil granulocytes [13]. Increased PRTN3 was shown to have a negative effect on the resolution of inflammation that causes immune system deregulation [14] The strikingly high PRTN3 expression that we observed in SARS-CoV-2 positive patients compared to negative group (over 29 fold, $p < 0.05$), may be the indicator of such phenomenon.

Multifunctional protease CTSG is also thought to be critically important in the maintenance of the delicate balance between tissue protection and destruction during inflammatory responses [15]. As a component of neutrophil proteolytic machinery CTSG regulates the inflammatory responses by stimulating the production of cytokines and chemokines, which are responsible for the activation and mobilization of immune cells to the site of pathogen and/or tissue damage [16,17]. CTSG activates metalloproteases and cleaves extracellular matrix proteins, contributing to neutrophil migration [18]. CTGS upregulation (more than 3 fold, $p < 0.05$) that we observed in SARS-CoV-2 patients, possibly is another indicator for abnormal neutrophil accumulation in nasopharynx.

ELANE has a physiological function as a powerful host defense, but is also known as one of the most destructive enzymes in the body. An overwhelming release of enzymatically active ELANE can cause local tissue injury [19]. Addition to that it is also reported ELANE can activate the spike (S) protein of coronaviruses and shift the viral entry to a low pH-independent route [20]. Therefore, the highly expressed ELANE ($\approx$3 fold, $p < 0.05$) that we observed in SARS-CoV-2 infected patients is supporting these findings.

SARS-CoV-2 infected group appeared to be highly expressing the AZU1 protein (heparin-binding protein/cationic antimicrobial protein of 37 kD) which is mobilized rapidly from emigrating polymorphonuclear leukocytes (PMN). Initially, this inactive serine protease was recognized for its antimicrobial effects. However, it soon became apparent that azurocidin may act to alarm the immune system in different ways and thus serve as an important mediator during the initiation of the immune response. Azurocidin, released from PMN secretory vesicles or primary granules, acts as a chemoattractant and activator of monocytes and

macrophages. The functional consequence is enhancement of cytokine release and bacterial phagocytosis, allowing for a more efficient bacterial clearance. Leukocyte activation by azurocidin is mediated via beta(2)-integrins, and azurocidin-induced chemotaxis is dependent on formyl-peptide receptors. In addition, azurocidin activates endothelial cells leading to vascular leakage and edema formation.

## 4.2. AG proteins and respiratory tract diseases

AG proteins' active role in neutrophil-associated lung inflammatory and tissue-destructive diseases has been reported [21]. Increased expressions of PRTN3, ELANE, and CTSG was reported in COPD patients [22]. In a mouse model study ELANE or PRTN3 was introduced to the trachea which caused tissue destruction and enlargement in airspace [23]. ELANE expression was also implicated in the impairment of host defense resulting in a decrease in mucociliary clearance of bacteria and also pathogens phagocytosis [24]. CD2, CD4, and CD8 can be cleaved on the surface of T-cells by ELANE and CTSG that dysregulate T-cell function [25]. It was reported that patients that lack alpha-1 antitrypsin (α1-Pi) which is the physiological inhibitor of PRTN3 and ELANE carries a high risk of developing emphysema [26]. Azurophil granules also carry Cathepsin D which was reported to be up-regulated in the pulmonary macrophages in a mouse model of cigarette smoking [27]. Severe COPD cases exhibited MPO positive cells as a signal of neutrophil activation [28]. On the other hand in a mouse model of influenza it was shown that the inflammation damage was reduced by the absence of MPO [29]. Regarding the high expression results of these protein markers we may suggest that during the progression of SARS-CoV-2 infection the same molecular mechanisms are most likely to be induced.

## 4.3. AG proteins and cytokine driven immune dysregulation

Among the AG proteins that we have identified, especially PRTN3, ELANE and CTSG were mostly associated with cytokine driven immune dysregulation. IL-32, a proinflammatory cytokine with four isoforms, cleavage by PRTN3 propagates cytokine activity and triggers IL-1beta, TNF-alpha, IL-6, and chemokines. It was argued that the targeted inhibition of PR3 or silencing of IL-32 by an inactive form of PRTN3 may halt the IL-32 driven immune dysregulation [30]. ELANE, CTSG, and PRTN3 can cleave pro-IL-1beta to bioactive IL-1beta [31]. Caspase-1/Interleukin-1 converting enzyme (ICE) cleaves proteins like precursors of the inflammatory cytokines interleukin 1β and interleukin 18 into their mature biologically active forms [32] and CTSG regulates Caspase-1 in this pathway [33]. ICE has an active role in cell immunity as an initiator of inflammatory response so once activated it triggers the formation of active IL-1beta which is secreted from the cell that induces inflammation in the neighboring cells [34]. It was recently reported that infection in COVID-19 patients with acute respiratory syndrome showed release of the pro-inflammatory cytokines like IL-1beta and IL-6 [35].

## 4.4. AG proteins and their role in NETosis

NETosis is a type of programmed cell death where neutrophil extracellular traps are formed [22]. NETs were identified in 2004 and they are often overlooked as drivers of severe pathogenic inflammation [36]. NETs have pathogen killing properties and include strands of DNA wrapped with histones and are enriched with neutrophil proteins like MPO, ELANE, PRTN3 and AZU1 [37] which were also present in our SARS-CoV-2 (+) samples with high amounts. Although NETosis is a very powerful mechanism fighting for the infection, the ability of NETs to damage tissues is well-documented in infection and sterile disease. NETs directly kill

epithelial and endothelial cells [11, 38], and excessive NETosis damages the epithelium in pulmonary fungal infection [12] and the endothelium in transfusion-related acute lung injury [39].

## 5. Conclusions

Through the available literature, we can see that the up-regulation of various proteins observed in the naso-oropharyngeal swab samples of COVID-19 patients is tightly interconnected with the immune response. The alterations of various proteins in SARS-CoV-2 infected patients' naso-oropharyngeal samples depict the molecular changes that govern the host antiviral defense system. The available literature for many respiratory diseases are very helpful in linking altered protein expressions to viral pathogenesis. Obtained data provided us an important view of SARS-CoV-2 molecular changes on the protein level in the infection site. Statistically significant protein alterations of PRTN3, MPO, ELANE, CTSG, and AZU1 dysregulation is important in the early phases of infection and may be targets for anti-SARS-CoV-2 therapeutics. Further research may show a link between the level of these proteins with disease severity and may be used as prognostic markers. Modulating the dysregulated proteins like PRTN3 or MPO may promote an anti-inflammatory response to alleviate SARS-CoV-2 symptoms. Also targeting NETs to dampen the out-of-control host response as a treatment may increase the survival rate by reducing the number of patients who require mechanical ventilation in ICU. We posit here that excess NETs may elicit the severe multi-organ consequences of COVID-19 via their known effects on tissues and the immune, vascular, and coagulation systems. Targeting NSPs and NETs in COVID-19 patients should therefore be considered by the biomedical community.

## Supporting information

**S1 Table. Protein quantification data.**
(XLS)

**S2 Table. Protein identification data.**
(XLS)

## Author Contributions

**Conceptualization:** Mete Bora Tuzuner, Mustafa Serteser, Ibrahim Unsal, Ahmet Tarik Baykal.

**Data curation:** Mete Bora Tuzuner, Hacer Nur Cakiroglu, Ahmet Tarik Baykal.

**Formal analysis:** Emel Akgun, Betul Sahin, Hacer Nur Cakiroglu, Ahmet Tarik Baykal.

**Investigation:** Mustafa Serteser, Ibrahim Unsal, Ahmet Tarik Baykal.

**Methodology:** Meltem Kilercik, Canan Kulah, Ibrahim Unsal, Ahmet Tarik Baykal.

**Project administration:** Ahmet Tarik Baykal.

**Resources:** Mustafa Serteser.

**Supervision:** Ahmet Tarik Baykal.

**Writing – original draft:** Emel Akgun, Mete Bora Tuzuner, Betul Sahin, Meltem Kilercik, Canan Kulah, Mustafa Serteser, Ibrahim Unsal, Ahmet Tarik Baykal.

**Writing – review & editing:** Ahmet Tarik Baykal.

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
