## [Decision Letter · Decision Letter 0]

29 Jul 2020

PONE-D-20-19154

ALTERED MOLECULAR PATHWAYS OBSERVED IN NASO-OROPHARYNGEAL SAMPLES OF SARS-CoV-2 PATIENTS

PLOS ONE

Dear Dr. Baykal,

Thank you for submitting your manuscript to PLOS ONE. After careful consideration, we feel that it has merit but does not fully meet PLOS ONE’s publication criteria as it currently stands. Therefore, we invite you to submit a revised version of the manuscript that addresses the points raised during the review process.

The data presented are interesting. However, the manuscript has several major flaws which require careful attention. All of these are outlined in the comments to the authors. I recommend the authors to fully address these comments.

We look forward to receiving your revised manuscript.

Kind regards,

Nades Palaniyar, MSc., PhD.

Academic Editor

PLOS ONE

Journal Requirements:

2. Please include a copy of Table 1 which you refer to in your text on page 6.

Reviewers' comments:

Reviewer's Responses to Questions

**Comments to the Author**

1. Is the manuscript technically sound, and do the data support the conclusions?

Reviewer #1: Partly

2. Has the statistical analysis been performed appropriately and rigorously? 

Reviewer #1: Yes

3. Have the authors made all data underlying the findings in their manuscript fully available?

Reviewer #1: Yes

4. Is the manuscript presented in an intelligible fashion and written in standard English?

Reviewer #1: No

5. Review Comments to the Author

Reviewer #1: Recent research clearly demonstrates that a delicate balance exists between protective immune responses that effectively clear SARS-CoV-2 infections with mild symptoms, versus overly exuberant immune responses that are likely to worsen disease outcome. Thus, understanding the nature of immune responses elicited by SARS-CoV-2 infection remains an important area of study. In this manuscript, Akgun et al. perform proteomic analysis of nasopharyngeal swabs from SARS-CoV-2 positive and negative patients. The authors report an enrichment in proteins associated with neutrophil degranulation/NETosis. The data itself is interesting however, the manuscript is poorly written and lacks several important details (including a critical data Table). These issues should be carefully addressed to improve the manuscript.

Major Comments:

1. The authors must provide more information about the participants in this study. This should include demographics (age, sex, etc.) as well as data pertaining to when swabs were taken (relative to onset of symptoms for example) and ideally, clinical severity. Were the SARS-CoV-2-negative participants healthy, or did they have some other ailment?

2. Table 1 (noted on line 134) appears to be missing from the manuscript file. This table contains the central data of the manuscript and its absence makes reviewing very difficult.

3. Supplementary tables should be called out in the text.

4. The discussion is too long and unfocused. For example, the authors spend considerable space discussing the role of NETosis in autoimmunity, which is completely tangential and should be removed. Instead, the authors should focus specifically on how their data fits in with reported characteristics of SARS-CoV-2 infections - specifically the role of neutrophils/NETs.

5. The discussion lacks balance. The authors seem to favor the hypothesis that their protein signature (or neutrophil degranulation/NETosis in general) is problematic, but provide not evidence to support that notion. It is certainly possible that dysregulated neutrophil responses could be pathogenic, but a balanced response might also be protective. Since the results presented by the authors is entirely descriptive, they should endeavor to provide a more balanced discussion of their interpretation.

6. The title of the manuscript is misleading. The authors should a difference between the proteomic profiles of infected vs. uninfected individuals. Thus, the fact that pathways are "altered" compared to SARS-CoV-2 negative participants is not surprising. A more descriptive title would be more appropriate. For example "Proteins associated with neutrophil degranulation are upregulated in nasopharyngeal swabs from SARS-CoV-2 patients"

Minor comments:

1. The manuscript needs to be thoroughly edited for English language/grammar.

6. PLOS authors have the option to publish the peer review history of their article (what does this mean?). If published, this will include your full peer review and any attached files.

Reviewer #1: No

---

## [Author Response · Author response to Decision Letter 0]

15 Sep 2020

PONE-D-20-19154

ALTERED MOLECULAR PATHWAYS OBSERVED IN NASO-OROPHARYNGEAL SAMPLES OF SARS-CoV-2 PATIENTS

PLOS ONE

Dear Dr. Palaniyar,

Thank you for the evaluation of our manuscript. We revised the manuscript with careful attention regarding to the comments of the reviewers and our responses to the comments were shared below. Hopefully this version of the manuscript will fully meet PLOS ONE’s publication criteria. 

We look forward for the decision of our revised manuscript.

Kind regards,

Ahmet Tarık Baykal, PhD.

Reviewer #1: Recent research clearly demonstrates that a delicate balance exists between protective immune responses that effectively clear SARS-CoV-2 infections with mild symptoms, versus overly exuberant immune responses that are likely to worsen disease outcome. Thus, understanding the nature of immune responses elicited by SARS-CoV-2 infection remains an important area of study. In this manuscript, Akgun et al. perform proteomic analysis of nasopharyngeal swabs from SARS-CoV-2 positive and negative patients. The authors report an enrichment in proteins associated with neutrophil degranulation/NETosis. The data itself is interesting however, the manuscript is poorly written and lacks several important details (including a critical data Table). These issues should be carefully addressed to improve the manuscript.

Major Comments:

1. The authors must provide more information about the participants in this study. This should include demographics (age, sex, etc.) as well as data pertaining to when swabs were taken (relative to onset of symptoms for example) and ideally, clinical severity. Were the SARS-CoV-2-negative participants healthy, or did they have some other ailment?

-We added age, gender and neutrophil count data to the “Study Population and Sample Collection” section of the manuscript. Patient selection criteria and conditions of the patients were also added to the same section.

2. Table 1 (noted on line 134) appears to be missing from the manuscript file. This table contains the central data of the manuscript and its absence makes reviewing very difficult.

-We added the table in the manuscript.

3. Supplementary tables should be called out in the text.

-Supplementary tables were cited in the results of the manuscript, under the “Label free proteomics” subsection. 

4. The discussion is too long and unfocused. For example, the authors spend considerable space discussing the role of NETosis in autoimmunity, which is completely tangential and should be removed. Instead, the authors should focus specifically on how their data fits in with reported characteristics of SARS-CoV-2 infections - specifically the role of neutrophils/NETs.

-The discussion part was rewritten. The role of NETosis in autoimmunity part was removed as recommended. We explained the possible dysregulation of neutrophil degranulation and NETosis caused by SARS-CoV-2 infection according to our findings with two subsection under the discussion part. 

5. The discussion lacks balance. The authors seem to favor the hypothesis that their protein signature (or neutrophil degranulation/NETosis in general) is problematic, but provide not evidence to support that notion. It is certainly possible that dysregulated neutrophil responses could be pathogenic, but a balanced response might also be protective. Since the results presented by the authors is entirely descriptive, they should endeavor to provide a more balanced discussion of their interpretation.

-The discussion part of the manuscript was reorganized according to the comments. We explained the role of each protein signature for the tissue injury and the abnormal inflammation, via relating our findings with the support of the current literature. 

6. The title of the manuscript is misleading. The authors should a difference between the proteomic profiles of infected vs. uninfected individuals. Thus, the fact that pathways are "altered" compared to SARS-CoV-2 negative participants is not surprising. A more descriptive title would be more appropriate. For example "Proteins associated with neutrophil degranulation are upregulated in nasopharyngeal swabs from SARS-CoV-2 patients"

- Title was changed to “Proteins associated with neutrophil degranulation are upregulated in nasopharyngeal swabs from SARS-CoV-2 patients” as per review suggestion. We thank the reviewer for the suggestion as it clearly summarizes the proteomic study. 

Minor comments:

1. The manuscript needs to be thoroughly edited for English language/grammar.

- The whole manuscript was checked and edited for English language/grammar.

---

## [Editor Report · Decision Letter 1]

18 Sep 2020

Proteins associated with neutrophil degranulation are upregulated in nasopharyngeal swabs from SARS-CoV-2 patients

PONE-D-20-19154R1

Dear Dr. Baykal,

We’re pleased to inform you that your manuscript has been judged scientifically suitable for publication and will be formally accepted for publication once it meets all outstanding technical requirements.

Kind regards,

Nades Palaniyar, MSc., PhD.

Academic Editor

PLOS ONE
---

## [Editor Report · Acceptance letter]

28 Sep 2020

PONE-D-20-19154R1 

Proteins associated with neutrophil degranulation are upregulated in nasopharyngeal swabs from SARS-CoV-2 patients 

Dear Dr. Baykal:

I'm pleased to inform you that your manuscript has been deemed suitable for publication in PLOS ONE. Congratulations! Your manuscript is now with our production department. 

Kind regards, 

on behalf of

Dr. Nades Palaniyar 

Academic Editor

PLOS ONE